



**Response of protonated, adduct, and fragmented ions in Vocus proton-**
**transfer-reaction time-of-flight mass spectrometer (PTR-ToF-MS)**
Fangbing Li[1], Dan Dan Huang[2], Linhui Tian[1], Bin Yuan[3], Wen Tan[4], Liang Zhu[4],
Penglin Ye[5], Douglas Worsnop[5], Ka In Hoi[1], Kai Meng Mok[1], Yong Jie Li[1]
[1]Department of Civil and Environmental Engineering, Department of Ocean Science
and Technology, and Centre for Regional Oceans, Faculty of Science and Technology,
University of Macau, Macau, China
[2]State Environmental Protection Key Laboratory of Cause and Prevention of Urban Air
Pollution Complex, Shanghai Academy of Environmental Sciences, Shanghai, China
[3]Institute for Environment and Climate Research, Jinan University, Guangzhou 510632,
China
[4]Tofwerk AG, Nanjing, China
[5]Aerodyne Research, Inc., Billerica, Massachusetts 01821, United States
*Correspondence to:* Yong Jie Li (yongjieli@um.edu.mo)





**Abstract**
Volatile organic compounds (VOCs) affect secondary pollutant formation via active
chemistry. Proton-transfer-reaction mass spectrometry (PTR-MS) is one of the most
important techniques to study the highly variable spatial and temporal characteristics
of VOCs. The response of protonated, adduct, and fragmented ions in PTR-MS in
changing instrument settings and varying relative humidity (RH) requires rigorous
characterization. Herein, dedicatedly designed laboratory experiments were conducted
to investigate the response of these ions for 21 VOCs, including 12 oxygenated VOCs
and two nitriles, using the recently developed Vocus PTR-MS. Our results show that
the focusing ion-molecule reactor (FIMR) axial voltage increases sensitivity by three
to four orders of magnitude but does not significantly change the fractions of protonated
ions. Reducing the FIMR pressure, however, substantially increases fragmentation.
Applying a high radio frequency (RF) amplitude radially on FIMR can enhance
sensitivity by one to two orders of magnitude without affecting the protonated ion
fractions. The change in big segmented quadrupole (BSQ) amplitude mainly affects
sensitivity and protonated ion fraction by modifying ion transmission. The relationship
between sensitivity and proton-transfer reaction rate constant is complicated by the
influences from both ion transmission and protonated ion fraction. The protonated ions
of most VOCs studied (19 out of 21) show less than 15% variations in sensitivity as RH
increases from ~5% to ~85%, except for some long-chain aldehydes which show a
positive RH variation of up to 30%. Our results suggest that the Vocus PTR-MS can
reliably quantify the majority of VOCs under ambient conditions with varying RH.
However, caution is advised for small oxygenates such as formaldehyde and methanol
due to their low sensitivity, as well as for long-chain aldehydes for their slight RH
dependence and fragmentation.





## 1 Introduction

Atmospheric volatile organic compounds (VOCs) affect atmospheric chemistry by forming secondary pollutants such as tropospheric $O_3$ (Shao et al., 2016) and secondary organic aerosols (SOA) (Shrivastava et al., 2017). In addition to their low mixing ratios (parts per billion by volume, ppbv, or even lower), the spatial and temporal variabilities of atmospheric VOCs pose another analytical challenge to the study of their atmospheric occurrence, sources, and fates. Mass spectrometric (MS) techniques based on ion-molecule reactions (IMR) (Španěl and Smith, 1996) or specifically proton-transfer reactions (PTR) (Hansel et al., 1995;Lindinger et al., 1998) in a selected ion flow tube (SIFT) have been developed to provide fast-responding measurements of VOCs. These techniques, especially the PTR-MS, have been widely used in VOC measurements in outdoor and indoor environments (Salazar Gómez et al., 2021;Sekimoto and Koss, 2021;Pagonis et al., 2019;Pleil et al., 2019;Claflin et al., 2021;Schripp et al., 2014;Jensen et al., 2021).

Quantification of VOCs (denoted as M) by PTR-MS relies heavily on their proton-transfer reactions with the hydronium ion $H_3O^+$ (R1). In early SIFT-MS studies where reagent ions include a multitude of $H_3O^+(H_2O)_n$ (n =0, 1, 2, 3…) ion series (Španěl and Smith, 2000), proton-transfer reactions with more hydrated (n $\geq$ 1) hydronium ions (R2a) are also important for species with proton affinity (PA) larger than water clusters. In addition, ligand switching reactions (R2b) and association reactions (R3, with N being $N_2$ or $O_2$) are also common, leading to $[MH + H_2O]^+$ instead of $MH^+$. Under these circumstances, the quantification of VOCs might be heavily influenced by water vapor concentration, or relative humidity (RH), of the sample. For instance, acetone concentrations in exhaled air were overestimated by 13% even using both protonated ($MH^+$) and water adduct ($[MH + H_2O]^+$) ions for quantification, when water vapor varied in the range of $(1 - 10) \times 10^{12}$ molecules cm$^{-3}$ (Španěl and Smith, 2000). A later study (Smith et al., 2001) showed that quantification of other oxygenated VOCs (OVOCs) such as ethyl acetate, diethyl ether, methanol, ethanol, and propanol by SIFT-MS also suffered from RH dependence to various degrees.

$$M + H_3O^+ \rightarrow MH^+ + H_2O \qquad\qquad (R1)$$

$$M + H_3O^+(H_2O)_n \rightarrow MH^+ + (H_2O)_{n+1} \qquad\qquad (R2a)$$



$$M + H_3O^+(H_2O)_n \rightarrow [MH + H_2O]^+ + (H_2O)_n \qquad (R2b)$$

$$MH^+ + H_2O + N \rightarrow [MH + H_2O]^+ + N \qquad (R3)$$

For PTR-MS that normally uses the $MH^+$ for quantification, RH dependence was
also widely reported. For instance, Warneke et al. (2001) reported that the sensitivity of
benzene in PTR-MS decreased significantly with the increase of RH, while Steinbacher
et al. (2004) suggested a slight decrease with the increase of RH. Quantification of
biogenic volatile organic compounds (BVOCs) was also reported to be slightly affected
by RH (Kari et al., 2018). The RH dependence stems from the change of reagent ion
distribution, i.e., among $H_3O^+$ and $H_3O^+(H_2O)_n$ (n $\geq$ 1), which can lead to
overestimation or underestimation of VOCs if such dependence is strong because
ambient RH is deemed highly variable. Therefore, RH-dependent calibrations for VOC
measurements using PTR-MS were normally recommended (de Gouw and Warneke,
2007;Inomata et al., 2008;Sinha et al., 2009;Vlasenko et al., 2010;Cui et al.,
2016;Michoud et al., 2018).
Another complication in VOC measurements using SIFT-MS or PTR-MS is that,
due to the nucleophilicity of the oxygen atom, protonated OVOCs would dehydrate,
forming fragmented ions (R4). This reaction often occurs in heavy alcohols, aldehydes,
and carboxylic acids (Španěl et al., 1997;Ŝpaněl and Smith, 1998;Hartungen et al.,
2004;Baasandorj et al., 2015).
$MH^+ \rightarrow [MH - H_2O]^+ + H_2O \qquad (R4)$
In addition, cleavage on the C-C bond of the protonated ion (R5) is also possible,
especially for alkyl-substituted VOCs under high-energy conditions (e.g., a high E/N
ratio, which is the reduced electric field parameter with E being the electric field and N
the number density of the gas in the drift tube).
$MH^+ \rightarrow [MH - C_xH_y]^+ + C_xH_y \qquad (R5)$
For instance, at an E/N ratio of 120 Townsend (Td), substituted monocyclic aromatic
compounds such as ethylbenzene and propylbenzene start to fragment into a benzenium
ion ($C_6H_7^+$) (de Gouw et al., 2003;Gueneron et al., 2015).
A newly designed focusing ion-molecule reactor (FIMR) was used for PTR-MS,
termed Vocus, and has been shown to have little RH dependence for the protonated ion
because of the high concentration of water vapor introduced into the drift tube



(Krechmer et al., 2018). The concentration of hydronium ion ($H_3O^+$) in the drift tube is
high enough to maintain at a constant level and dominate over other side reactions,
thereby minimizing the RH dependence for VOC measurement. Yet, the formation of
adduct ions and fragmented ions in Vocus PTR-MS as a function of RH has not been
fully scrutinized, hindering a complete understanding of the ion chemistry in the Vocus
PTR-MS and potential cross interference when measuring ambient air with complex
VOC mixtures. Herein, we conducted experiments on the effects of instrumental
settings and RH variations on the quantification of 21 VOCs, including 12 OVOCs and
2 nitriles, using a Vocus PTR-MS. Response of protonated ions ($MH^+$), adduct ions
($[MH + H_2O]^+$), and fragmented ions ($[MH - H_2O]^+$ or $[MH - C_xH_y]^+$) of these VOCs
was investigated as a function of instrumental setting and RH. Results are interpreted
based on the PA values and/or proton-transfer reaction rate constants ($k_{ptr}$). Some
caveats on using the Vocus PTR-MS to measure VOCs, especially OVOCs, are also
provided.

## 2 Methodology

### 2.1 Instrument settings

Experiments were performed with a Vocus proton-transfer-reaction time-of-flight
mass spectrometer (PTR-ToF-MS, Vocus 2R, TOFWERK AG, Thun, Switzerland),
hereinafter referred to as Vocus. The Vocus consists of (i) a discharge ion source, (ii) a
focusing ion-molecule reactor (FIMR), (iii) a big segmented quadrupole (BSQ), (iv) a
series of direct current (DC) optics that further focus and accelerate the primary beam
(PB), and (v) a time-of-flight (ToF) mass analyzer (Krechmer et al., 2018). The ion
source is a plasma discharge composed of two conical surfaces. Water vapor is supplied
by purging 20 to 30 mL of milli-Q water and is ionized by plasma discharge. The
reagent ions pass through a ring offset from the central axis so that the photons
generated by the discharge cannot enter. The drift tube was improved by replacing the
stacked ring electrodes of the traditional PTR-MS with a FIMR, which is a glass tube
with a resistive coating on the inner surface and a quadrupole with a radio frequency
(RF) electric field applied. The FIMR increases ion transmission by a factor of 7 to 9
and sensitivity by more than one order of magnitude (Krechmer et al., 2018). Moreover,
the mean kinetic energy of $H_3O^+$ is increased by three times, and the formation of more
hydrated hydronium ions is reduced, suppressing RH dependence for most VOCs



132 measured (Krechmer et al., 2018). Meanwhile, the mean kinetic energy of VOCs

133 measured is not significantly increased, thereby minimizing fragmentation (Krechmer

134 et al., 2018). The ToF mass analyzer offers a mass resolving power of 12,000 at a mass-

135 to-charge ratio ($m/Q$) of 107 Thomson (Th).

136  In our experiments, sample air was drawn into the instrument using 0.5 m long

137 perfluoroalkoxy (PFA) Teflon tubing of ~0.5 m length and $1/4''$ outer diameter, with a

138 flow rate of 0.5 L·min$^{-1}$. Most of the sample air was directed to the exhaust, while the

139 actual flow into the FIMR was around 0.15 L·min$^{-1}$. In typical experiments, the drift

140 tube was operated at a pressure of 2.0 mbar and a temperature of 373.15 K. The axial

141 and radial voltages were normally set to be 625 and 500 V, respectively, unless stated

142 otherwise.

143  We also performed experiments by varying the instrument settings such as FIMR

144 axial voltage (V) and FIMR pressure (p), both of which affect the E/N ratio, as well as

145 RF and BSQ amplitudes to investigate how protonated, adduct, and fragmented ions

146 respond to those changes. These experiments were performed under dry (RH ~5%)

147 conditions, and the concentrations were approximately 12 ppbv for most VOCs (except

148 for β-caryophyllene at about 1.2 ppbv). The instrument settings were varied by: 1)

149 changing the FIMR axial voltage from 260 to 700 volts, 2) changing the pressure in

150 FIMR from 1.5 to 3.5 mbar; 3) changing the RF amplitude from 13 to 500 volts (with

151 p of 2.0 or 3.5 mbar); 4) changing the BSQ voltage from 50 to 300 volts (with p of 2.0

152 or 3.5 mbar). The other instrument settings were fixed as the default values while

153 changing the tested ones. Specifically, RF amplitude was at 500 volts and BSQ

154 amplitude was at 300 volts when changing E (i.e., V) and N (i.e., p), and an E/N ratio

155 of 142 Td was used when changing RF and BSQ amplitudes (Table S1).

156 **2.2 Experimental setup**

157  The VOCs (Table S2 and Figure 1) in mixtures from two cylinders were separately

158 delivered to the dilution and/or RH control setup (Figure S1). Dilution air was generated

159 from a zero-air generator (Environics series 7000, Environics Analytics Group Ltd.,

160 Canada). Gas cylinder I (Apel-Riemer Environmental Inc., US, valid for 12 month)

161 contains mainly hydrocarbons, while gas cylinder II (Linde Gases, US, valid for 12

162 month) contains mainly OVOCs and nitriles. Most VOCs in the cylinders are at

163 approximately 1000 ppbv, except for β-caryophyllene that is at approximately 100 ppbv.





Table S2 shows their CAS numbers, *m/Q* values of the protonated ions (MH$^+$), as well
as PA and $k_{ptr}$ values. According to their functional groups, the 21 VOCs are grouped
into 9 categories, and Figure 1 shows their structures. Note that although n-butanal and
methyl ethyl ketone are isomers, they are in different cylinders and measured separately,
thus will not interfere with each other.

169        Dry experiments were performed by diluting the VOCs from the cylinders with

dry zero air with 7 concentrations from 0 to approximately 22 ppbv (or approximately
2 ppbv for β-caryophyllene). For each concentration step, measurements lasted for
about half an hour for gas cylinder I with hydrocarbons but about two hours for gas
cylinder II with OVOCs and nitriles. Other measures such as minimizing the length of
the Teflon tube at the inlet (less than 30 cm) were also undertaken to allow fast
establishment of equilibrium-state concentrations for OVOCs. In RH-dependent
experiments (Figure S1), dilution was made by RH-conditioned air produced from a
humidity generator (OHG-4, Owlstone, US). The accuracy of the RH sensor (RH-USB
Probe, Omega) is within 4% of RH. Nine RH ramping steps from ~5% to ~85% with
approximately 10% intervals were used, and the VOCs were set with 4 concentrations
from 0 to approximately 12 ppbv (or approximately 1.2 ppbv for β-caryophyllene).
After the initial equilibration of 0.5 hours under dry conditions (RH ~5%), each RH
ramping lasted for 15 min. Triplicate experiments were performed for the highest
concentration.



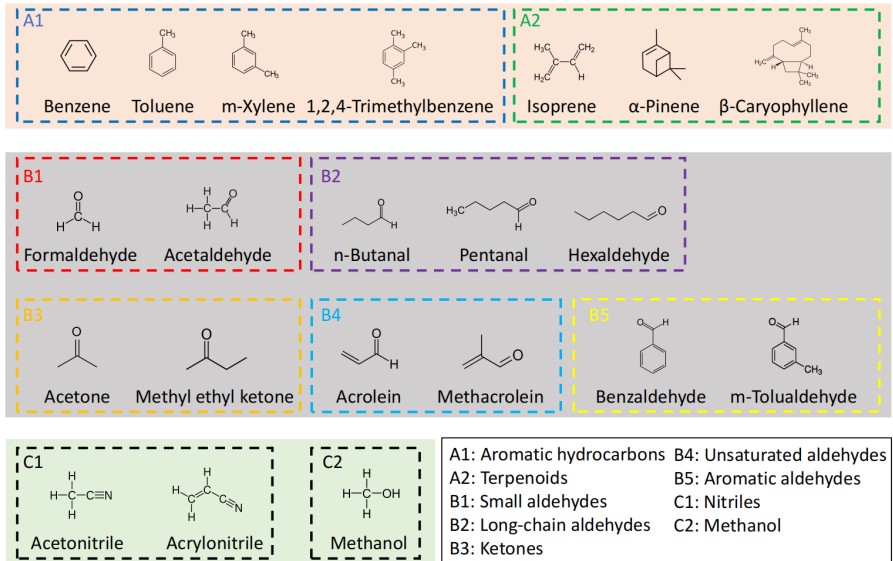

Figure 1. Names, structures, and grouping of the 21 VOCs in this study. These VOCs were prepared in two gas standard cylinders (I and II) with mixing ratios of ~1000 ppbv (~100 ppbv for β-caryophyllene), balanced by $N_2$. Acetaldehyde and acetone are present in both gas standard cylinders, with concentrations differing by <5%. n-Butanal and methyl ethyl ketone are isomers but are in different gas standard cylinders.

**2.3 Data analysis**

The Vocus data were analyzed with the manufacturer-supplied software package Tofware (v3.2.3) based on Igor Pro (Wavemetrics). Peak fitting was performed using Tofware routines and the measured $m/Q$ values of the pronated ions ($MH^+$) are shown in Table S2 together with their exact $m/Q$ values. In addition to $MH^+$, we also looked for adduct ions ($[MH + H_2O]^+$), and fragmented ions ($[MH – H_2O]^+$ or $[MH – C_xH_y]^+$), since those ions are also anticipated for VOCs, especially OVOCs, in PTR-MS measurements (Pagonis et al., 2019). Although the VOCs were in mixtures and showed ensemble mass spectra, we constructed the mass spectrum for each VOC by plotting their identified adduct or fragmented ions alongside the protonated ions, as shown in Figures S2-S4 in the SI. Their percentage contributions are presented in Figure 2 and numerically in Table S4.



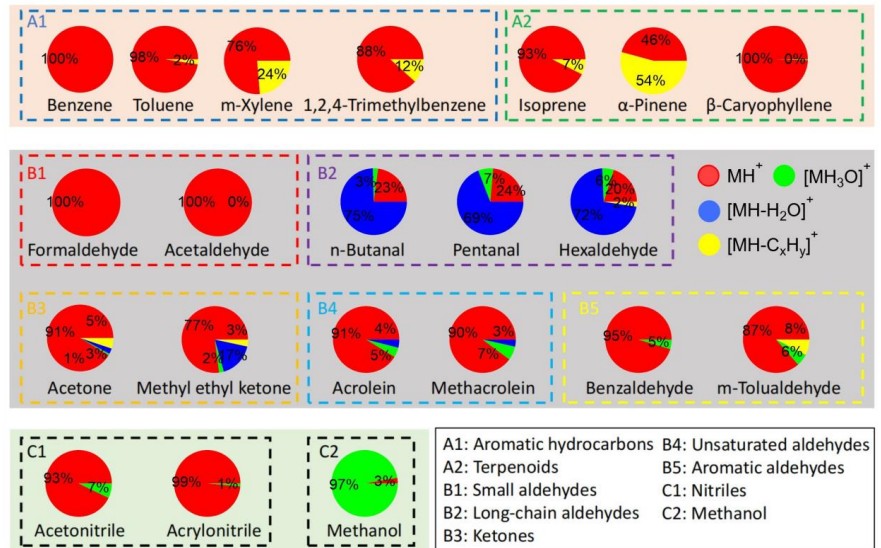

Figure 2. The average signal percentages of protonated, adduct, and fragmented ions at a concentration of ~12 ppbv (~1.2 ppbv for β-caryophyllene).

For both dry and RH-dependent experiments, from the 1-minute averages from 1-Hz datasets, the last five points were averaged to obtain stable signals. Figure S5 shows the time series of selected hydrocarbons (groups A1 and A2, aromatic hydrocarbons and terpenoids, Figure S5a) and OVOCs (groups B2 and B5, long-chain and aromatic aldehydes, Figure S5b) during dry experiments with concentration stepping. Figure S6 shows the time series of protonated ions (MH$^+$), adduct ions ([MH + H$_2$O]$^+$), and fragmented ions ([MH – H$_2$O]$^+$ and/or [MH – C$_x$H$_y$]$^+$ for n-butanal, pentanal and hexaldehyde, whose adduct and fragmented ions contributed substantially to the total signals (Figure 2 and Table S4).

Similar to RH-dependent experiments, the duration of each instrument setting experiment was 15 min. The signal intensities and the ratio of MH$^+$, [MH + H$_2$O]$^+$ and [MH – H$_2$O]$^+$ and/or [MH – C$_x$H$_y$]$^+$ to all for two typical hydrocarbons (α-pinene and 1,2,4-trimethylbenzene) and two OVOCs (acetone and hexaldehyde) as the axial voltage and pressure (which both affects the E/N ratio) in the FIMR, RF amplitude, and BSQ amplitude varied are shown in Figures S7 to S10. The reduced electric field parameter (E/N ratio) was estimated by comparing the signal fraction of fragment of α-pinene in Materić et al. (2017), in which detailed examination on fragment signal fraction at different E/N ratio was performed (Figure S11).



### 2.4 Sensitivity estimation

The formation of the protonated ion $MH^+$ via reaction R1 is desirable for quantification of VOCs, which is described by the kinetics of the proton-transfer reaction (de Gouw and Warneke, 2007; Yuan et al., 2017):

$$[MH^+] = [H_3O^+]_0 \ (1 - e^{-k[M]\Delta t}) \quad \text{(Eq. 1)}$$

where $[MH^+]$ is the number concentration of the protonated ion, $[H_3O^+]_0$ that of the initial hydronium ion, $k$ is the rate constant of R1 ($k_{ptr}$), $[M]$ is the number concentration of the target VOCs in the sample air, and $\Delta t$ is the reaction time in the drift tube. Two conditions allow simplification of Eq. 1 to Eq. 2 below for easy quantification of VOCs: 1) The term $k[M]\Delta t$ is much smaller than 1, such that R1 can be considered essentially first-order; and 2) $H_3O^+$ is not significantly depleted and remain more or less constant after the drift tube. Compared with traditional ion sources, the Vocus ion source produces sufficient $H_3O^+$ (Krechmer et al., 2018). Ambient levels of ppbv (or less) for $[M]$ ($\sim 10^{10}$ molecule cm$^{-3}$) generally fulfill such requirements, given that $k_{ptr}$ is on the order of $10^{-9}$ cm$^3$ molecule$^{-1}$ s$^{-1}$ and $\Delta t$ of $10^{-4}$ s (Ellis and Mayhew, 2014). Therefore,

$$[MH^+] = [H_3O^+]k[M]\Delta t \quad \text{(Eq. 2)}$$

where $[H_3O^+]$ is the mixing ratio of hydronium ions after the drift tube (i.e., being detected in the mass spectrometer). Then,

$$[M] = \frac{I_{MH^+}}{I_{H_3O^+}} \frac{1}{k\Delta t} \quad \text{(Eq. 3)}$$

where $I_{MH^+}$ and $I_{H_3O^+}$ are signal intensities of the protonated ion and the hydronium ion, respectively.

In general, the sensitivity (S) of PTR-MS for quantification of VOCs is defined as the ratio between the signal intensity $I_{MH^+}$ normalized by $10^6$ cps (counts per second) of $I_{H_3O^+}$ and 1 ppbv ($10^{-9}$ mol mol$^{-1}$) of VOCs, i.e.:

$$S = \frac{\frac{I_{MH^+}}{I_{H_3O^+}} \times 10^6}{\frac{[M]}{N} \times 10^9} \quad \text{(Eq. 4)}$$




where N is the number density of air in the drift tube. The sensitivity S is thus expressed as a normalized signal per ppbv, having a unit of ncps ppbv$^{-1}$. Combining Eq. 3 and Eq. 4 yields,

$$S = 10^{-3} \times N\Delta t \times k \text{ (Eq. 5)}$$

where $10^{-3} \times N \Delta t$ is specific to the instrumental settings. Eq. 5 dictates that S should have a linear relationship with the proton-transfer reaction rate constant ($k_{ptr}$) if the instrument settings are fixed and can be utilized to predict S if $k_{ptr}$ values are known (Ellis and Mayhew, 2014).

In reality, however, quantification of VOCs using MH$^+$ from PTR-MS measurements is complicated by 1) formation of adduct (e.g., with H$_2$O) and fragmented (e.g., dehydration) ions, and 2) discriminated transmission for MH$^+$ ions with different $m/Q$ values (de Gouw and Warneke, 2007;Yuan et al., 2017). The fraction of MH$^+$ in all related ions ($f_{MH^+}$) and the relative transmission efficiency ($T_{MH^+}/T_{H_3O^+}$) are used to account for these two effects, respectively:

$$S = 10^{-3} \times N\Delta t \times \frac{T_{MH^+}}{T_{H_3O^+}} \times f_{MH^+} \times k \text{ (Eq. 6)}$$

In our study, the sensitivity is expressed as the slope of signal intensity (in counts per second, cps) vs. concentration (in ppbv), having a unit of cps ppbv$^{-1}$ (Figure S12). Signal normalization to H$_3$O$^+$ (ncps) was not adopted because the signal of H$_3$O$^+$ ($m/Q$ = 19 Th) was substantially suppressed with low transmission (see below) for those ions with small $m/Q$ values (but too high intensities) to minimize ion currents.

**3 Results and Discussion**

**3.1 Effects of instrumental settings on the ion signals**

An increase of the E/N ratio from 48 to 142 Td manifested by the increases of the FIMR axial voltage (V, with front from 226 to 666 volts and back keeps at 34 volts) led to drastic increases of MH$^+$ signal intensity by three to four orders of magnitude for all VOCs studied (Figure 3a). Such increases were also observed for adduct and fragmented ions (Figures S7 - S10), albeit to different extents. It has been shown that increasing the axial voltage in the FIMR can substantially increase protonated ion signals, which is due mainly to three reasons (Krechmer et al., 2018;de Gouw and



Warneke, 2007). First, a high FIMR axial voltage can accelerate the ions and thus
reduce their residence time, thereby preventing diffusional loss. Second, the high
voltage in FIMR significantly increases the concentration of all reagent ions (Figure
S13). Lastly, at high voltage, reactions between some OVOCs (such as acetone) and
$H_3O^+$ leads to more protonated ions ($MH^+$) instead of adduct ions ($[MH + H_2O]^+$)
through ligand switching reaction (R2b). The last effect was believed to be less
significant for species that do not readily react with $H_3O^+(H_2O)_n$ (such as aromatics and
terpenoids).

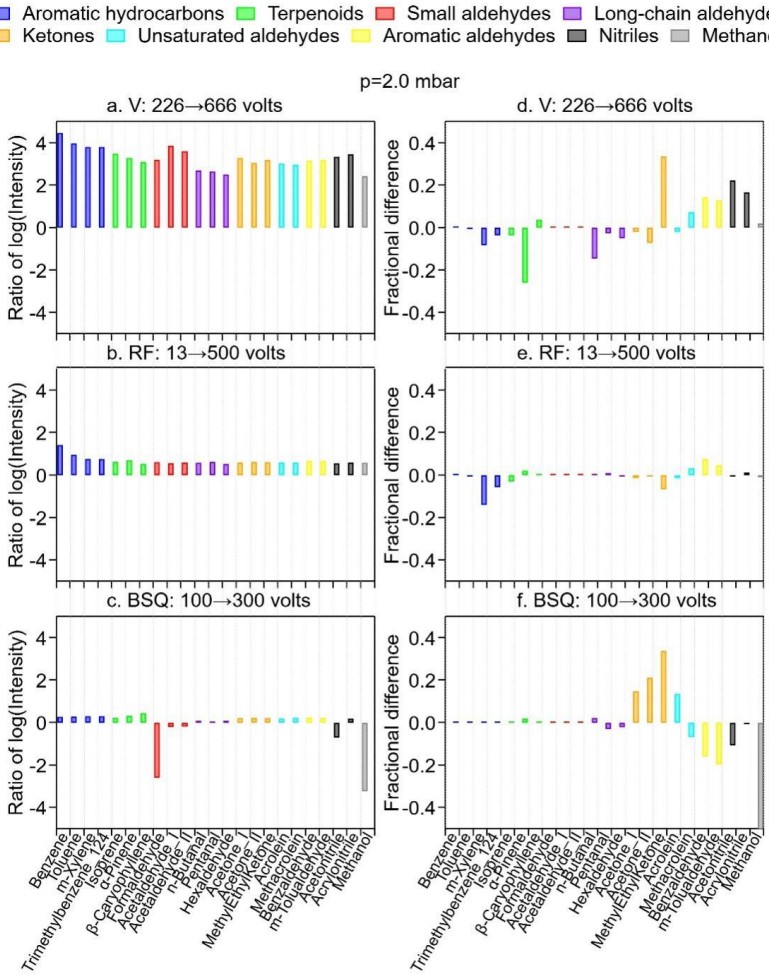


Figure 3. The ratio of the logarithm of intensity (panels a, b, and c) and the difference
of fractional signal of the protonated ion ($MH^+$) among all ions (panels d, e, and f),





when changing axial voltage (V) or FIMR pressure (p) (panels a and d), RF amplitude
(panels b and e), and BSQ amplitude (panels c and f). The ratios were taken after taking
the logarithm of the signal intensities of $MH^+$ at the right-hand side of the instrument
setting (after the arrow) to that at the left-hand side of the instrument setting stated in
the panel label; likewise, the fractional differences are the fractions of the $MH^+$ signal
among the protonated, fragmented, and adduct signals under these two instrumental
settings.
To investigate whether the desired $MH^+$ is indeed more favorably formed by
suppressing R2b under high axial voltages, we plot the differences in the signal
fractions of $MH^+$ between axial voltages of 666 volts and 226 volts (Figure 3d). The
results show that about one-third of the 21 VOCs do not have significant differences in
the signal fractions: most VOCs in this one-third have the $MH^+$ as the sole or
dominating ion observed (*cf.*, Figure 2). Meanwhile, there are about one-third showing
negative differences (i.e., decreasing $MH^+$ fractions) of up to 0.2, including the two
hydrocarbons and two OVOCs shown in panel d of Figures S7 – S10. The remaining
one-third show positive differences (i.e., increasing $MH^+$ fractions) of up to 0.3, mainly
for unsaturated or aromatic aldehydes, as well as nitriles (Figure 3d). A closer
inspection of the fractional changes as axial voltage increases for acetone (Figure S9d)
and hexaldehyde (Figure S10d) reveals that the fractions of both $MH^+$ and $[MH + H_2O]^+$
decrease, while those of fragmented ions ($[MH – H_2O]^+$ and/or $[MH – C_xH_y]^+$) increase.
The relative decreases of the signal fractions of $MH^+$ (8% for acetone and 51% for
hexaldehyde) are, however, much lower than those of $[MH + H_2O]^+$ (57% for acetone
and 80% for hexaldehyde). These observations suggest that while fractions of both $MH^+$
and $[MH + H_2O]^+$ decrease, the decreases of the adduct ion ($[MH + H_2O]^+$) are more
significant, supporting the third reason that relatively more $H_3O^+$ (instead of
$H_3O^+(H_2O)_n$) at higher axial voltages to react with these OVOCs. However, as the
voltage increases, all ion signals are increasing (Figure S7a-S10a). This observation
illustrates that the ion acceleration and diffusion prevention should be the primary
reason for signal enhancement at high axial voltages. Nevertheless, the signal fractions
of the $MH^+$ do not change substantially (within 30%) as the FIMR axial voltage
increases, making quantification reliable even for species with high signal contributions
from adduct and fragmented ions (e.g., long-chain aldehydes, group B2, *cf*., Figure 2).



The increase of E/N ratio by decreasing FIMR pressure from 3.5 to 1.5 mbar
increases signal intensities (Figure S14a) of $MH^+$ by less than one order of magnitude,
or even decreases those for some OVOCs such as long-chain aldehydes. The changes
in signal fractions of $MH^+$ (Figure S14d), on the other hand, are more than those when
changing axial voltages, especially for ketones, unsaturated aldehydes, aromatic
aldehydes, and nitriles. The increase of pressure in the PTR reactor also favors the
formation of reagent clusters $H_3O^+(H_2O)_n$, which leads to the formation of adduct ions
(Wang et al., 2020). For OVOCs acetone and hexaldehyde, the fractions of $MH^+$ do
increase when FIMR pressure was decreased from 3.5 to 2.5 mbar (E/N ratio from 162
to 95), which is accompanied by significant decreases of the adduct ion $[MH + H_2O]^+$
(Figure S9j and Figure S10j). This observation suggests less formation of adduct ions
at lower FIMR pressures. Further decrease of FIMR pressure to 1.5 mbar, however,
results in slight decreases of $MH^+$ fractions, in lieu of increases of fragmented ions [MH
$– H_2O]^+$ and $[MH – C_xH_y]^+$ (Figure S9j and Figure S10j); for hydrocarbons α-pinene
and 1,2,4-trimethylbenzene (Figure S7j and Figure S8j) that only have $MH^+$ and  [MH
$– C_xH_y]^+$, continuous decreases of $MH^+$ fractions and increases of $[MH – C_xH_y]^+$ are
observed for the whole range of FIMR pressure tested (3.5 to 1.5 mbar). A recent study
using Vocus PTR-MS to measure organic peroxides also observed that less fragmented
ions were formed under higher FIMR pressure (Li et al., 2022), presumably due to the
efficient transfer of excess kinetic energy by frequent collisions at higher pressures. A
medium FIMR pressure of 2.0 mbar was chosen to have relatively low fractions of both
adduct ions ($[MH + H_2O]^+$) and fragmented ions ($[MH – H_2O]^+$ and $[MH – C_xH_y]^+$).
The radial RF electric field in the FIMR is unique for the Vocus PTR-MS, which
can (1) collimate the ions towards the central axis (especially heavier ions) and (2)
increase the kinetic energy of the ions (especially for lighter ions) (Krechmer et al.,
2018). These effects led to 1 to 1.5 orders of magnitude higher signals for $MH^+$ at 2.0
mbar FIMR pressure (Figure 3b) and 1.5 to 2 orders of magnitude at 3.5 mbar (Figure
S14b) when the RF amplitude was changed from 13 to 500 volts. The additional
enhancement of signal intensity at a higher FIMR pressure (i.e., 3.5 mbar as compared
to 2.0 mbar) can be attributed to a longer residence time of the reagent ions (Krechmer
et al., 2018). The more pronounced increase of kinetic energy for lighter ions (e.g.,
$H_3O^+$) than heavier ions [i.e., clusters $H_3O^+(H_2O)_n$] might imply the favorable
formation of the protonated ion $MH^+$ rather than adduct ions. The fractions of $MH^+$ for





different RF amplitudes do not change significantly (within ±0.2) either at 2.0 mbar
(Figure 3e) or 3.5 mbar (Figure S14e). This observation thus suggests that adding the
RF can increase signal intensities by 1 – 2 orders of magnitude but does not affect the
fractional signal for MH$^+$, making it beneficial for accurate quantification.

357        The BSQ amplitude above 100 volts does not change the signal intensities

significantly (Figure 3c and Figure S14c, as well as Figures S7– S10). The BSQ ion
guide provides a high-pass band filter to reduce the number of ions (thus signal intensity)
of low $m/Q$ values (especially for reagent ions, H$_3$O$^+$, with high ion currents),
preventing the fast degradation of the microchannel plate (MCP) detector (Krechmer et
al., 2018). This bandpass filter leads to lower ion transmission efficiency (<1) for ions
with smaller $m/Q$ values, which is discussed below. Therefore, the signal reduction
when the BSQ amplitude increased from 100 to 300 volts is more obvious for small
analytes such as formaldehyde, acetonitrile, and methanol (Figure 3c and Figure S14c).
For other analytes whose fragmented ions have $m/Q$ values of less than 60 Th, the signal
fractions of MH$^+$ would also be affected (Figure 3f and Figure S14f). For example, the
intensities of fragmented ions [MH – H$_2$O]$^+$ (CH$_3$CCH$_2^+$, $m/Q$ = 41 Th) and [MH –
C$_x$H$_y$]$^+$ (CH$_3$CO$^+$, $m/Q$ = 43 Th) for acetone had substantial decreases when BSQ
amplitude was higher than 200 volts (Figure S9c and Figure S10). The protonated ion
MH$^+$ (CH$_3$COCH$_3$H$^+$, $m/Q$ = 59 Th) and adduct ion (CH$_3$COCH$_3$H$_3$O$^+$, $m/Q$ = 77 Th),
however, remained less unaffected. This effect leads to noticeable changes in signal
fractions of MH$^+$ (maximum 0.4) for small analytes such as ketones, unsaturated
aldehydes, nitriles, as well as methanol (Figure 3f and Figure S14f) as the BSQ
amplitude changes from 100 to 300 volts.
**3.2 Sensitivity and transmission of protonated ions**

377        We calculated the sensitivity and transmission of protonated ions (MH$^+$) of the 21

VOCs studied when the instrument was under the optimized conditions. Table 1 shows
the sensitivities (cps ppbv$^{-1}$), as slopes of MH$^+$ signals vs. mixing ratios (average value
from 0 to 22 ppbv at dry condition, except for β-caryophyllene to 2 ppbv) and the limit
of detection (LOD, 3σ). Panels a and b in Figure 4 show the sensitivity vs. $k_{ptr}$ for all
21 VOCs, while panels c and d show the transmission efficiencies calculated from the
division of the sensitivity vs. $k_{ptr}$ ratio for each VOC by the slope fitted in Figure 4a and
4b.



Most VOCs had sensitivities above 1000 cps ppbv$^{-1}$, except 1) formaldehyde (A2-
1) and methanol (C2-1), whose MH$^+$ ions have $m/Q$ values much lower than 60 Th; and
2) β-caryophyllene (A2-3) that came with a very low concentration range. In addition
to its low $m/Q$ values that limit the transmission, the backward reaction is also an
important reason for the low sensitivity. For instance, formaldehyde has a low PA value
(712.5 kJ mol$^{-1}$) that is not much higher than that of water (691.0 kJ mol$^{-1}$) and has been
shown to have a high tendency of backward reaction of R1 (Inomata et al.,
2008;Vlasenko et al., 2010;Warneke et al., 2011). The two compounds in group B5
(aromatic aldehydes), benzaldehyde and m-tolualdehyde, had the highest sensitivities
of >12000 cps ppbv$^{-1}$. This might be due to their high PA values (>830 kJ mol$^{-1}$), which
are among the highest except those of terpenoids (Table S2).

Table 1. Sensitivity (slope), intercept, and limit of detection (LOD) based on 3 standard
deviations (σ). Results were obtained from measurements of 0 – 22 ppbv for all VOCs
except for β-caryophyllene (up to ~2 ppbv).

| Group[a] | Name | Label | Sensitivity (cps ppbv$^{-1}$) | 3σ LOD (pptv), 5 s |
|---|---|---|---|---|
| A1 | Benzene | A1-1 | 2596 | 35 |
| | Toluene | A1-2 | 5724 | 2 |
| | m-Xylene | A1-3 | 8669 | 3 |
| | 1,2,4-Trimethylbenzene | A1-4 | 8951 | 1 |
| A2 | Isoprene | A2-1 | 2140 | 16 |
| | α-Pinene | A2-2 | 4046 | 2 |
| | β-Caryophyllene | A2-3 | 723 | 1 |
| B1 | Formaldehyde | B1-1 | —[c] | —[c] |
| | Acetaldehyde | B1-2 | 2096[b] | 283 |
| B2 | n-Butanal | B2-1 | 1114 | 343 |
| | Pentanal | B2-2 | 1465 | 63 |
| | Hexaldehyde | B2-3 | 1595 | 35 |
| B3 | Acetone | B3-1 | 9932[b] | 127 |
| | Methyl ethyl ketone | B3-3 | 9636 | 51 |
| B4 | Acrolein | B4-1 | 7224 | 16 |
| | Methacrolein | B4-2 | 6090 | 13 |
| B5 | Benzaldehyde | B5-1 | 16089 | 1 |
| | m-Tolualdehyde | B5-2 | 12893 | 1 |
| C1 | Acetonitrile | C1-1 | 1511 | 6 |
| | Acrylonitrile | C1-2 | 9275 | 1 |





| | | | | |
|---|---|---|---|---|
| C2 | Methanol | C2-1 | —c | —c |

Notes:
a, A1: aromatic hydrocarbons, A2: terpenoids, B1: small aldehydes; B2: long-chain aldehydes, B3:
ketones, B4: unsaturated aldehydes, B5: aromatic aldehydes, C1: nitriles, C2: methanol;
b, average value from gas standard cylinders I and II;
c, low sensitivity and high LOD due to low transmission.

Figure 4. Sensitivity as a function of $k_{ptr}$ for (a) MH$^+$ and (b) all ions. Linear fitting was
performed only for MH$^+$ sensitivity. Species included in the fitting were those with $m/Q$
value > 60 Th (Table S2) and signal percentage of MH$^+$ ($f_{MH^+}$) > 75% (Table 1). The
grey-shaded area is bounded by 0.5 × Slope and 2 × Slope. The fitted curves in panel b
are the same as in panel a and are for reference only. Panels c and d are transmission
curves for MH$^+$ and all ions. The sigmoidal curve for MH$^+$ (same for c and d) was fitted
from species except for β-caryophyllene, α-pinene, n-butanal, pentanal, and
hexaldehyde whose fragmentation was significant. Note that only the $m/Q$ values of





MH$^+$ was used in the x axis of panel d, which does not consider the differences in $m/Q$
values of adduct and fragmented ions.

417        It was shown that the sensitivities for different VOCs in PTR-MS can be calculated

from the kinetics of the proton-transfer reactions (Warneke et al., 2003;Sekimoto et al.,
2017;Cappellin et al., 2012). For Vocus, Krechmer et al. (2018) also pointed out that
the relationship between sensitivity and $k_{ptr}$ can be established and used to calculate the
sensitivity for other compounds. We herein compare the relationship between
sensitivity and $k_{ptr}$ (Figure 4a). In our data, the uncertainties for sensitivity were
conservatively taken as the maximum percentage uncertainty (5.3%) of fitted slopes.
Values of $k_{ptr}$ were calculated as averages of both modeled and experimental results
found from literature (Table S3), with uncertainties propagated from an estimated
percentage error of 15% for both modeled (Zhao and Zhang, 2004) and experimental
values. The anticipated linear relationship of sensitivity vs. $k_{ptr}$ is not easily visible,
most likely due to the formation of fragments/adducts for some VOCs and low
transmission efficiencies for others. However, a relatively improved linear relationship
was found if we limit the VOCs to 1) $m/Q$ values for MH$^+$ > 60 Th, and 2) a fraction of
MH$^+$ ion in all ions (including adduct/fragmented ions) larger than 75% (*cf*, Figure 2).
With these limitations, the fitted linear line gives a slope of $(2.8 \pm 0.3) \times 10^{12}$ cps ppbv$^-$
$^1$ molec s cm$^{-3}$, approximately 38% lower than that $[(4.5 \pm 0.4) \times 10^{12}$ cps ppbv$^{-1}$ molec
s cm$^{-3}$] of Krechmer et al. (2018). Pearson's R (R$_{Pr}$) is 0.77. A grey area is also shown
by two lines of 2 × slope and 0.5 × slope, which includes approximately half (ten) of
the VOCs studied. Those that fall out of the grey area to the lower region are mainly
compounds in groups B2 (long-chain aldehydes, purple), C1 and C2 (nitriles or
methanol, black), B1 (small aldehydes, red), and A2 (terpenoids, green). Using the total
signals of all ions (protonated, adduct, and fragmented), Figure 4b shows the
improvements for compounds in group B2 only, while others (especially those in C1,
C2, and B1) do not move up to the grey area.

442        We also calculated the transmission efficiencies (Figure 4c and 4d) from the

division of the sensitivity vs. $k_{ptr}$ ratio for each VOC by the slope fitted in Figure 4a. It
is shown that compounds in groups B1, C1, and C2 are mainly on the rising range of
the sigmoidal curve, while the three long-chain aldehydes in group B2 (purple) are well
below the curve if only MH$^+$ ions were used; these three long-chain aldehydes move up
to transmission > 0.5 when all ions are considered (Figure 4d). Of particular interest is





the compound α-pinene, whose transmission was < 0.5 when only the MH$^+$ ion was
used, but it increases to about 1.5 when the fragmented ion was considered. Note that
in Figure 4d, the $m/Q$ value of MH$^+$ was used in the x axis for the summed ion signals,
which might lead to certain bias as the fragmented and adduct ions have different $m/Q$
values. Nevertheless, the above analysis suggests that both the formation of
adduct/fragmented ions and transmission affect the relationship between sensitivity and
$k_{ptr}$, which needs precaution when using predicted sensitivity directly from $k_{ptr}$. In
addition, one needs to be cautious about the prediction of transmission efficiency with
$m/Q$ greater than 150.

**3.3 RH dependence of ion signals**

One of the most important reasons for RH dependence is that the distribution of
reagent ions might vary with ambient RH, especially when the abundance of H$_3$O$^+$ in
the PTR reactor is not high. While the Vocus has been shown to have abundant enough
H$_3$O$^+$ (Krechmer et al., 2018), whether it can substantially minimize RH dependence
for most VOCs deserves scrutinization. Figure 5a-h shows the relative sensitivity,
defined as the relative change of sensitivity of MH$^+$ ion vs. VOC concentration under
different RH conditions. Among the nine groups of VOCs studied, seven groups show
almost flat relative sensitivities within the RH range of ~5% to ~85% (298 K), with the
exceptions of long-chain aldehydes (group B2) that show increasing sensitivities as RH
increases (Figure 5d) and methanol (group C2) showing large variations (Figure 5h).
Some other compounds, such as β-caryophyllene (A2-3, Figure 5b) and formaldehyde
(B1-1, Figure 5c) also show either relatively large uncertainties or fluctuations, which
can be ascribed to their low intensities (*cf.*, Table 1).

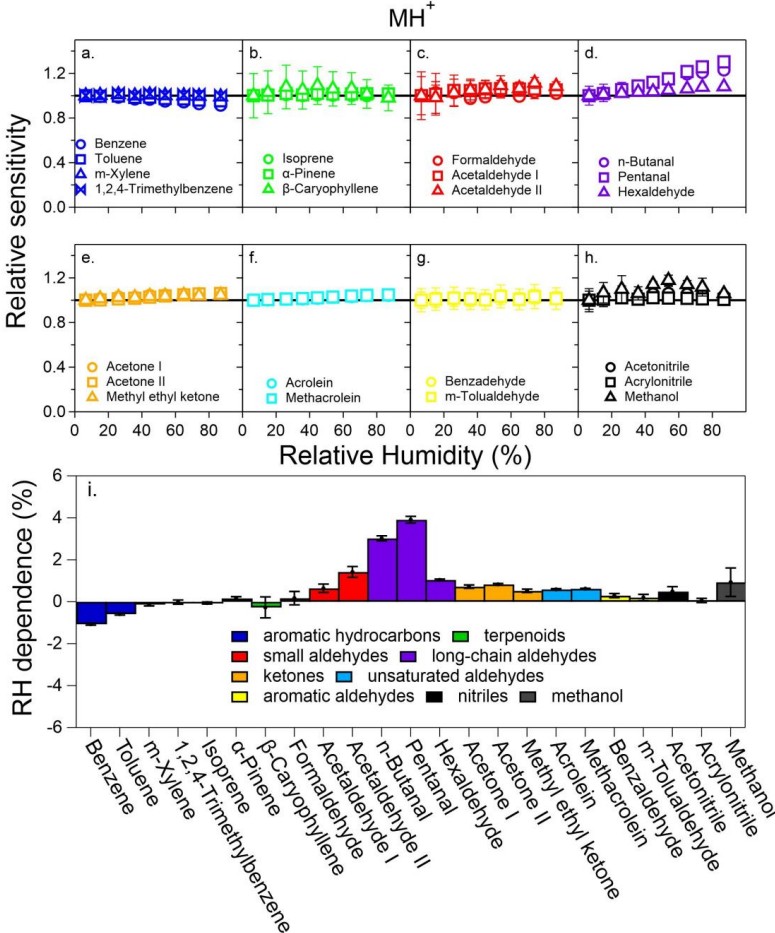


Figure 5. The dependence of the MH$^+$ signals on RH for the VOCs studied. Panels a-h:
the relative sensitivity was calculated as the slope (Sensitivity) under all conditions to
that at the dry (RH<5%) condition. Panel i: the percent change of relative sensitivity
per 10% RH increase. See Figures S15-S17 for other ion signals.
Figure 5i shows the RH dependence of the MH$^+$ ions, defined as the percentage
change of sensitivity per 10% RH increase, for all 21 VOCs studied. Aromatic
hydrocarbons (group A1) show negative RH dependence as in previous studies using
PTR-MS with a drift tube (Warneke et al., 2001;Steinbacher et al., 2004). While
previous studies reported decreases in benzene sensitivity by 16 – 56% from dry to
humid conditions (up to 100% RH), our results show a decrease of less than 1.1% per
10% RH increase (i.e., <11% in the whole RH range) with a somewhat narrower RH




range (up to ~85% RH). In addition, two out of the three terpenoids (group A2) also
show slightly negative RH dependence, and the rest one (α-pinene) shows very small
positive RH dependence (Figure 5i). These hydrocarbons (aromatics and terpenoids) in
groups A1 and A2 have relatively low $k_{ptr}$ values (mostly $<2.5 \times 10^{-9}$ cm$^3$ molec$^{-1}$ s$^{-1}$,
Table S3), and their RH dependence shows a fairly good correlation with the PA value
(Figure 6a, purple circles and squares). This observation suggests that there might be a
thermodynamic reason behind the noticeable decrease of sensitivity for hydrocarbons
such as benzene as RH increases. Since hydrocarbons such as benzene and toluene do
not readily react with $H_3O^+(H_2O)$ (Warneke et al., 2001), R1 is the main reaction to
form $MH^+$. As water vapor concentration increases at high RH, the reverse reaction of
R1 might be important for compounds with low PA and low $k_{ptr}$ values.

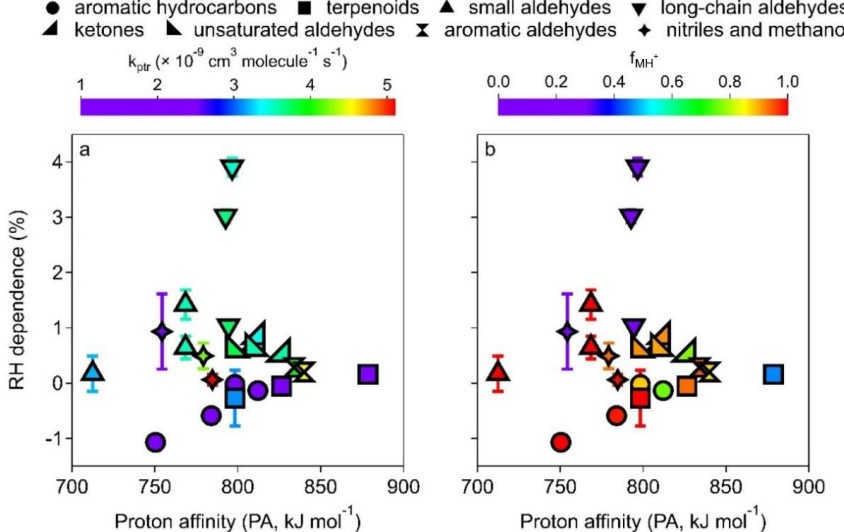


Figure 6: The RH dependence of $MH^+$ ion plotted against PA, color-coded by (a) $k_{ptr}$
and (b) $f_{MH^+}$.
Long-chain aldehydes (group B2) have the largest RH dependence of 1-4%
positive deviation per 10% RH increase for the $MH^+$ ions. The RH dependence of the
$[MH + H_2O]^+$ ions (Figure S15g) is even much higher (1.4–8.5% positive deviation per
10% RH increase). Interestingly, the trends of RH dependence for the $MH^+$ ions and
that for the $[MH + H_2O]^+$ ions for long-chain aldehydes are exactly opposite (Figure 5i
and Figure S15g); that is, pentanal > n-butanal > hexaldehyde. The reason behind this



observation is out of the scope of this study. The dominating $[MH - H_2O]^+$ ions for
long-chain aldehydes (Figure 2), however, show much less (±1% per 10% RH increase)
RH dependence (Figure S16i). Other carbonyl compounds (groups B1, B3, B4, and B5)
also show positive deviations as RH increases (less than 1.5% increase in sensitivity
per 10% RH increase), albert to various degrees (Figure 5i). Similar to long-chain
aldehydes, their $[MH + H_2O]^+$ ions also show a large positive deviation (Figure S16g),
and $[MH - H_2O]^+$ ions show little RH dependence (Figure S16e). These carbonyl
compounds have medium PA values (Figure 6a) except formaldehyde. If we exclude
formaldehyde (with an extremely low PA value, the up triangle to the far left in Figure
6b) and long-chain aldehydes (low percentages of $MH^+$ ions, <25%, down triangles in
Figure 6b), the RH dependence of other carbonyl compounds shows a slightly
decreasing trend of RH dependence vs. PA values (Figure 6b, up triangles, left triangles,
right triangles, and double triangles). These observations might hint on the relationships
between the RH dependence of carbonyl compounds and reagent ion distribution as
well as reaction direction for R1-R4, which is different from those for pure
hydrocarbons (groups A1 and A2). Finally, the RH dependence of $MH^+$ for compounds
in groups C1 (nitriles) and C2 (methanol) is within +1% per 10% RH increase (Figure
5i).
Overall, in the whole RH range studied (~5% to ~85%), the RH dependence of
$RH^+$ ions for the 21 VOCs studied is less than 30%, with most compounds (except
group B2, long-chain aldehydes) less than 15%. For $[MH + H_2O]^+$ ions (mainly for
carbonyl compounds), strong RH dependence was observed (Figure S15), being 1.4 –
8.5% per 10% RH increase, or 8.9 – 63.2% from ~5% to ~85% RH. The dehydrated
ions ($[MH - H_2O]^+$), however, show the smallest RH dependence (±1% per 10% RH
increase) among all the ions (Figure S16). Fragmented ions with decarbonization ($[MH$
$- C_xH_y]^+$) show mainly negative RH dependence, generally less than 3% per 10% RH
increase (Figure S17).

**4 Conclusions**

We investigated the response of protonated, adduct, and fragmented ions of 21
atmospherically relevant VOCs in a Vocus PTR-MS as instrument setting and RH
condition vary. For the two ways of increasing the E/N ratio, increasing the FIMR axial
voltage can substantially (by three to four orders of magnitude) increase sensitivity but




does not change the fractions of the MH$^+$ ions (mostly within 30%); reducing the FIMR
pressure, however, does not enhance sensitivity much but can lead to more substantial
fragmentation. Therefore, a high FIMR axial voltage of 600 – 700 volts and a medium
pressure of around 2.0 mbar are recommended. Increasing the RF amplitude of FIMR
can increase sensitivity by 1 to 1.5 orders of magnitude at 2.0 mbar and 1.5 to 2 orders
of magnitude at 3.5 mbar, and it does not change the MH$^+$ ion fractions (within 20%).
Therefore, a high RF amplitude of 500 V is recommended. Increasing the BSQ
amplitude does not increase the sensitivity much but changes the MH$^+$ ion fractions of
small ions substantially by changing the transmission efficiency. The choice of this
instrument setting mainly relies on what ions (i.e., those reagent ions with too high an
abundance) one wants to filter out. Our choice is at 300 V, which gives a 50%
transmission at about 55 Th.
The relationship between sensitivity and $k_{ptr}$ is strongly affected by two factors: 1)
whether the MH$^+$ ion has a high transmission efficiency, and 2) whether the MH$^+$ ion
is the dominating ion. If so, a fairly good correlation ($R_{Pr} = 0.77$) was observed for the
VOCs studied. The transmission curve is also more reasonably resembling the sigmoid
function only if all the ions (protonated, adduct, and fragmented) are considered. The
low transmission efficiencies of formaldehyde and methanol result in extremely low
sensitivities of these two small OVOCs, although a low PA value is another reason for
the former.
As RH increases from ~5% to ~85%, the MH$^+$ ions for 19 out of the 21 VOCs
studied have sensitivity variation of less than 15%, but long-chain aldehydes have
positive RH dependence of up to 30%. The RH dependence of [MH + H$_2$O]$^+$ ions for
long-chain aldehydes is stronger, while that of the dominating [MH – H$_2$O]$^+$ ions is
limited. Therefore, the signal distributions among protonated, adduct, and fragmented
ions are also affected by RH variation. Together with their relatively high background
signals (especially for n-butanal, Figure S5b), quantification of long-chain aldehydes
in the ambient environment using Vocus requires special attention. It is also worth
noting that hydrocarbons generally show slight negative RH dependence, probably due
to their relatively low $k_{ptr}$ values, although such RH dependence does not affect
quantification significantly; their RH dependence has a fairly good correlation with
their PA values, hinting a thermodynamic reason behind this trend.



**Competing interests**

At least one of the (co-)authors is a member of the editorial board of Atmospheric
Measurement Techniques

**Acknowledgments**

This work was supported by funding support from the Science and Technology
Development Fund, Macau SAR (File No. FDCT 0031/2023/AFJ) and a multiyear
research grant (No. MYRG2022-00027-FST) from the University of Macau.



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
