# Peer review of "Response of protonated, adduct, and fragmented ions in Vocus proton- transfer-reaction time-of-flight mass spectrometer (PTR-ToF-MS)"

_EGUsphere, 2024_

## Author Comment (AC1)

We sincerely thank the editor and reviewers for the valuable comments and suggestions, which are very helpful in improving our manuscript. We herein provide the point-by-point response and the changes made to the original manuscript according to the comments and suggestions. The response is in the indent and blue, and the revised text is in the indent and green. The line numbers mentioned in the response correspond to the revised manuscript unless otherwise specified.

Reviewer:

This study investigated the response of VOC product ions in the Vocus proton transfer-reaction time-of-flight mass spectrometer (PTR-ToF-MS) by manipulating instrument settings and varying relative humidity (RH). The findings demonstrated the reliable quantification capability of Vocus PTR-MS for a wide range of VOCs under ambient conditions with varying RH, while noting some underestimation for small oxygenates and long-chain aldehydes. The manuscript is well-written and have significance findings for future applications in the environmental atmospheric field. However, there are several issues that require addressing prior to publication.

Response: We appreciate the positive comments from the reviewer and the response to specific comments are provided below.

Major Comments:

1. The quantification of VOCs using PTR-MS primarily relies on their proton transfer reactions with $H_3O^+$. However, it is also necessary to consider the reaction involving $H_3O^+(H_2O)_n$ (where n = 1, 2, 3...) for species with a higher proton affinity (PA) than that of water clusters. Therefore, it is crucial to conduct all experiments in an environment where the reaction with $H_3O^+$ dominates and to identify the key parameters (E/N, FIMR, RF, BSQ…) that primarily minimize the influence of $H_3O^+(H_2O)_n$.

Response: Yes, indeed the distribution of reagent ions is important for reactions occurring in PTR-MS (protonation or other side reactions). In Vocus PTR-MS, the hydronium ion ($H_3O^+$, m/Q = 19) is so much dominating that an instrument setting

(BSQ) is deliberately tuned to filter out much of this low m/Q ion to protect the detector, but its abundance should be at least 10 times that of the next hydrated one ($H_3OH_2O^+$, m/Q = 37) (Krechmer et al., 2018). With a PA value of 808 kJ mol$^{-1}$ for the water dimer, $(H_2O)_2$ (Goebbert and Wenthold, 2004), $H_3OH_2O^+$ might react with some VOCs with PA values lower than that, e.g., benzene and toluene in group A1, small and long-chain aldehydes in groups B1 and B2, as well as nitriles and methanol in groups C1 and C2 in our study. Yet, the overwhelmingly dominating $H_3O^+$ in the reagent ion distribution led us to believe that protonation is the major reaction leading to ionization for most of the VOCs studied.

On the other hand, although we could not determine the actual abundance of hydronium ion ($H_3O^+$, m/Q = 19) in our experiments, we observed that the ratio of $H_3O^+$ to $H_3O^+(H_2O)_n$ (where n = 1 and 2) did not change very significantly (Figure R1.1 below, and Figure S13 in the revised SI) when we changed the instrument setting (except for BSQ in which m/Q discrimination was the reason behind). This observation suggests that when we changed the instrument settings, the dominance of $H_3O^+$ should not be affected.

[Figure]

Figure R1.1. The signal intensities of $H_3O^+$, $H_3OH_2O^+$, and $H_3O(H_2O)_2^+$ as conditions

varied.

2. The experiment utilized two gas cylinders containing mixed standard gases. So how to make sure that the results of the adduct/fragmented ions in Figure 2 are not interfered by different species in the mixed standard gases as the reactions occurring in VOCUS (R1-R5) are considerably complex.

Response: We constructed the mass spectra (Figures S2-S4) for each tested VOC by: 1) looking for the protonated ions and potential adduct and fragmented ions according to possible reactions of R1-R5; 2) comparing the signal variations of the suspected adduct and fragmented ions with that of the protonated ion (co-varying) to confirm that they are actually from the target VOC; and 3) expanding the search on possible fragmented ions, especially for $[MH – C_xH_y]^+$ in which x can be more than 1 (e.g., for α-pinene), with reported fragments from literature for PTR-MS analysis of VOCs.

Interference does occur for one typical case. The protonated signals $(MH^+)$ of acrolein and methacrolein (m/Q = 57 and 71, respectively) overlap with the fragmented ions $([MH – C_xH_y]^+)$ of n-butanal and pentanal, respectively (with x = 1, i.e., losing a methyl group). From the structurally analogous hexaldehyde, long-chain aldehydes should have a fragmented ion $[MH – C_xH_y]^+$ as that of hexaldehyde, but the signal percentages should be very small (see Figure 2, ~2% for hexaldehyde). Therefore, fragmented ions $([MH – C_xH_y]^+)$ of n-butanal and pentanal have little effect on the signal of $MH^+$ of acrolein and methacrolein, and the signals at m/Q = 57 and 71 are considered directly to belong to $[MH]^+$ of acrolein and methacrolein, respectively. In that case, the fragmented ions $[MH – C_xH_y]^+$ for n-butanal and pentanal cannot be obtained here, and it is assumed to be of a low signal percentage as that of hexaldehyde (~2%). In addition, for isomers n-butanal and methyl ethyl ketone, they are in two different cylinders and will not interfere with each other (as stated in L212 in the manuscript).

3. The findings regarding methanol present a puzzling scenario. Figure 2 shows that the majority of product ions from methanol are fragmented ions of [MH+H2O]+ (m/Q=51), accounting for 97% of the total. However, Figure 4

suggests that the sensitivity and transmission of methanol remain unchanged when considering all product ions. It is well-established that methanol can be effectively analyzed using traditional PTR and has been widely utilized in atmospheric analysis. Therefore, it is imperative to determine whether the reduced sensitivity of protonated methanol in VOCUS is primarily due to lower transmission or the prevalence of $[MH+H2O]+$.

Response: The dominating contribution of the adduct ion $[MH + H_2O]^+$ ($m/Q = 51$) for methanol is due primarily to the low transmission efficiency of the protonated ion $MH^+$ (see Figure 4c, approaching zero for methanol at $m/Q = 33$). With our BSQ setting of 300 volts, adduct ion $[MH + H_2O]^+$ ($m/Q = 51$) for methanol should have a transmission efficiency of approximately 50%. If we take into account of ion transmission and "scale" the signals of $[MH + H_2O]^+$ and $MH^+$ back to 100% transmission efficiency, the signal intensity of $MH^+$ can be as high as 1400 cps/ppbv, higher than that of $[MH+H_2O]^+$ (~ 90 cps/ppbv). We made a note on this in L551-554 of the manuscript together with formaldehyde (which has the problems of both low transmission and low PA value).

Minor Comments:

1.  Please ensure the form of symbols is written consistently in Section 4.2.

We assumed that for Section 3.2 as Section 4 is the Conclusions. We have double checked the consistency of the symbols used throughout the manuscript.

2.  Line 202. The label of the green color in Figure 2 should be consistently expressed as [MH+H2O]+.

Revised as below in Figure R1.2.

[Figure]

Figure R1.2. Revised Figure 2 in the main text.

3. Lines 492-493. "the reverse reaction of R1 might be important for compounds with low *PA* and low *kptr* values" needs to be cited with relevant references.

A reference of reverse reaction has been cited here.

L500: (Inomata et al., 2008).

4. Lines 414-415. The title and content are reversed in Figures 4 c and d. It seems only the m/Q values of MH+ are used in the x axis of panel c.

We are sorry for the confusion here. Actually, both panels c and d use the same set of values for x axis, which are the m/Q values of the MH$^+$ ions. But what is shown in panel d is the transmission by taking into account of all ions, instead of just MH$^+$ as in panel c. That is, the difference between these two panels lies in the data in the y axis, instead of x axis.

To avoid confusion, we have modified the caption of Figure 4 into:

L418 and L424: Panels c and d are transmission curves from MH$^+$ only and from the sum of all ions, respectively. …. Note that only the m/Q values of MH$^+$ were used in the x axes of both panels c and d, although panel d contains information of adduct and fragmented ions that have m/Q values different from that of MH$^+$.

5. Line 522. What does "RH+ ions" mean?

Revised.

L522: MH$^+$

6. There are many expressions of "drift tube". I suggest replacing the "drift tube" with the FIMR, as the "drift tube" is the reaction chamber in traditional PTR and it has been optimized in VOCUS.

Revised. The component used to refer to PTR is retained as drift tube, and the Vocus component is referred to as FIMR.

L99, L100, L138, L225, L237, L241, L246, and L255: FIMR.

References:

Goebbert, D. J., and Wenthold, P. G.: Water Dimer Proton Affinity from the Kinetic Method: Dissociation Energy of the Water Dimer, European Journal of Mass Spectrometry, 10, 837-845, https://doi.org/10.1255/ejms.684, 2004.

Inomata, S., Tanimoto, H., Kameyama, S., Tsunogai, U., Irie, H., Kanaya, Y., and Wang, Z.: Technical Note: Determination of formaldehyde mixing ratios in air with PTR-MS: laboratory experiments and field measurements, Atmospheric Chemistry and Physics, 8, 273-284, https://doi.org/10.5194/acp-8-273-2008, 2008.

Krechmer, J., Lopez-Hilfiker, F., Koss, A., Hutterli, M., Stoermer, C., Deming, B., Kimmel, J., Warneke, C., Holzinger, R., Jayne, J., Worsnop, D., Fuhrer, K., Gonin, M., and de Gouw, J.: Evaluation of a New Reagent-Ion Source and Focusing Ion–Molecule Reactor for Use in Proton-Transfer-Reaction Mass Spectrometry, Anal Chem, 90, 12011-12018, https://doi.org/10.1021/acs.analchem.8b02641, 2018.

---

## Author Comment (AC2)

We sincerely thank the editor and reviewers for the valuable comments and suggestions, which are very helpful in improving our manuscript. We herein provide the point-by-point response and the changes made to the original manuscript according to the comments and suggestions. The response is in the indent and blue, and the revised text is in the indent and green. The line numbers mentioned in the response correspond to the revised manuscript unless otherwise specified.

Reviewer:

This manuscript describes a series of tests performed on the newly developed Vocus PTR-MS to characterize the instrument response to humidity and various internal settings (voltages and pressures) for 21 commonly measured volatile organic compounds. The authors show that sensitivities and ion distributions for most species showed little variation with humidity, and provide recommendations for instrument settings to maximize sensitivities. The manuscript is well written, straightforward, easy to follow, and provides useful benchmarks that instrument users and developers will certainly find helpful. I would recommend publication following clarification of a few minor points:

Response: We appreciate the positive comments from the reviewer and the response to specific comments are provided below.

1. It should be mentioned somewhere which species are in which standard tank, and some evidence needs to be provided that the various fragments of each compound in an individual tank do not overlap with each other in any way that could obfuscate the results. Were these compounds ever tested outside the tanks to check that?

Response: Thanks for the suggestion. We have now included a new entry (Cylinder No.) in Table 1 (P17) to show the species in each cylinder.

L159 and L160: Table 1.

L405: The cylinder numbers for the VOCs studied are also shown here

As for the potential interference, we carefully compared the anticipated protonated ions, fragmented ions, and adduct ions as expected from reactions R1-R5 (also referring to literature for dealkylation for R5, e.g., for α-pinene). There is one such case. The protonated signals ($MH^+$) of acrolein and methacrolein (m/Q = 57 and 71, respectively) overlap with the fragmented ions ($[MH - C_xH_y]^+$) of n-butanal and pentanal, respectively (with x = 1, i.e., losing a methyl group). From the structurally analogous hexaldehyde, long-chain aldehydes should have a fragmented ion $[MH - C_xH_y]^+$ as that of hexaldehyde, but the signal percentages should be very small (see Figure 2, ~2% for hexaldehyde). Therefore, fragmented ions ($[MH - C_xH_y]^+$) of n-butanal and pentanal have little effect on the signal of $MH^+$ of acrolein and methacrolein, and the signals at m/Q = 57 and 71 are considered directly to belong to $[MH]^+$ of acrolein and methacrolein, respectively. In that case, the fragmented ions $[MH - C_xH_y]^+$ for n-butanal and pentanal cannot be obtained here, and it is assumed to be of a low signal percentage as that of hexaldehyde (~2%). We have included a few sentences in L215 to clarify that.

L215-L220: Some dealkylated fragments ($[MH - C_xH_y]^+$) of long-chain aldehydes (e.g., n-butanal and pentanal) might overlap with the protonated ions ($[MH]^+$) of unsaturated aldehydes (i.e., acrolein and methacrolein). Yet since the intensities of the former are expected to be low by analogy with that of hexaldehyde (~2%, Figure 2), the ions at those m/Q values are only considered as the protonated ions of unsaturated aldehydes (the latter).

In addition, n-butanal and methyl ethyl ketone are isomers. But they are in different cylinders and will not interfere with each other (as stated in L166 in the manuscript).

As shown in the response to the 2nd comment of Reviewer 1, we used a few measures to find the fragmented and adduct ions of a specific VOC, and reconstruct the mass spectrum (shown in Figures S2-S4). We did not do all the 21 compounds individually, but we did used a permeation/diffusion tube device to generate toluene. The mass

spectra of toluene from the mixed-VOC cylinder (reconstructed) and from the individual VOC are shown below in Figure R2.1.

[Figure]

Figure R2.1. Mass spectra of toluene from (a) mixed VOC cylinder and (b) from individual VOC.

2. How long were the tanks allowed to equilibrate on the instrument before ramping concentrations, humidity, and other variables? And was it ever demonstrated that the wait times of 30 minutes for one tank and 120 minutes for the other (L 171-173) were sufficient for stabilisation? From Figure S5 it appears that some species' signals are still creeping up by the end of the step. How much error could this introduce in the calibrations and the determinations of their sensitivities to instrument parameters?

Response: For most VOCs, especially hydrocarbons, the signal can attain stability within 10 minutes. For oxygenated VOCs (OVOCs), such as benzaldehyde and m-tolualdehyde, it does take longer (Figure S5). That is the reason why we prolonged the equilibration time for cylinder II, which contains mainly OVOCs and nitriles (Table 1). Even for a shorter equilibration time of 30 min for cylinder I, the overlapping species acetone and acetaldehyde showed deviations of only less than ±1.5% in sensitivity.

Therefore, we believe that for OVOCs in cylinder II, the equilibration time of 120 min is sufficient to reach a relatively stable signal for quantification.

We have included in the revised manuscript the following descriptions on this.

L172: The 2-hour stabilization time for cylinder II, which contained mainly OVOCs and nitriles, should be sufficient because even with half-an-hour stabilization time for cylinder I, the overlapping species acetone and acetaldehyde showed deviations of less than ±1.5% in sensitivity.

L410: with a deviation of less than ±1.5%.

3. Similarly, was it ever demonstrated that 15 minute steps of humidity (L 181-182) were sufficient to allow for signal stabilisation at each step?

Response: In doing the RH-controlled experiments, the concentrations of the VOCs were fixed at certain desired levels first. Then the RH of the experimental setup was ramped up and down, as shown in Figure S6. The RH within the experimental setup can fluctuate, but the standard deviations of RH change are normally less than 5%. This difference is in general smaller than the RH intervals we set for stepping (e.g., panels a-h in Figure 5), and therefore allows us to draw conclusions from the general trend of RH increasing and decreasing.

4. L 484: "the rest one" should be "the other one"

Revised.

L493: other.